# Working Conditions and Well-Being of School Nurses in Spain: Impact on Job Satisfaction and Professional Quality of Life

**DOI:** 10.3390/healthcare13030323

**Published:** 2025-02-04

**Authors:** José Antonio Zafra-Agea, Estel·la Ramírez-Baraldes, Eduard Maldonado-Manzano, Núria Obradors-Rial, Antònia Puiggrós-Binefa, Ester Colillas-Malet

**Affiliations:** 1Department of Nursing, Faculty of Health Sciences at Manresa, University of Vic–Central University of Catalonia, Av Universitària 4-6 cp, 08242 Manresa, Spain; 2Department of Nursing, Faculty of Nursing, Physiotherapy, and Podiatry, University of Seville, c/Avenzoar 6, 41009 Seville, Spain; 3CAP Verdaguer, Primary Care and Community Management Baix Llobregat, Catalan Institute of Health, University Institute for Primary Care Research (IDIAP Jordi Gol), 08007 Barcelona, Spain; 4Research Group on Epidemiology and Public Health in the context of Digital Health (Epi4Health), Institute of Research and Innovation in Life and Health Sciences of Central Catalonia (IRIS-CC), 08242 Vic, Spain; 5Intensive Care Unit, Althaia University Health Network, C/ Dr. Joan Soler 1-3, 08243 Manresa, Spain; 6Research Group on Transformative Innovation and Simulation (GRITS), Institute of Research and Innovation in Life and Health Sciences of Central Catalonia (IRIS-CC), 08242 Vic, Spain

**Keywords:** school nursing, job satisfaction, professional quality of life, work factors, educational settings

## Abstract

**Background:** School nurses play a crucial role in promoting student health, addressing issues such as substance use, mental health, physical health, violence, and sexual health. However, their job satisfaction has been understudied, particularly in relation to the challenges they face. **Objective:** This study evaluates the job satisfaction and professional quality of life among school nurses and nurses working in educational settings in Spain. **Methods:** A descriptive cross-sectional study was conducted (November 2023–February 2024) with 553 nurses from various regions of Spain, using the Font-Roja job satisfaction and CPV-35 professional quality of life questionnaires. **Results:** Our results showed significant differences in job satisfaction between nurses in healthcare and educational settings, influenced by factors such as workload and institutional support. Most participants (97.5% women, median age ~40 years) had less than 5 years of experience. The lack of a dedicated nurse in half of the schools highlights disparities in service provision. **Discussion:** This study emphasizes the need for supportive policies and work environments to improve the well-being and job satisfaction of nurses in school settings.

## 1. Introduction

School nurses in Spain are healthcare professionals dedicated to promoting the physical and emotional well-being of students. Their role is not limited to working directly within educational institutions; it also includes nurses who provide school health services in community/public health settings and private organizations [1]. These professionals possess a general nursing education obtained through a university degree, and many have further enhanced their qualifications with specialized master’s degrees in school, community, or public health nursing. Their diverse work environments enable them to manage a wide range of responsibilities, from providing immediate emergency care to promoting healthy habits and managing chronic conditions such as type 1 diabetes and asthma [2,3,4]. For example, school nurses monitor the blood glucose levels of students with diabetes, administer insulin when necessary, and educate both students and staff on proper disease management. In the case of asthma, they help students identify and avoid triggers, administer rescue medications, and develop individualized action plans to manage asthma exacerbations during school hours [2,3,4].

However, despite their essential role, school nurses in Spain may face challenges related to job satisfaction and quality of life due to the specific demands of their work environment [5,6,7,8].

Job satisfaction in nursing, and specifically in school nursing, is a concept that has been studied in various contexts. Generally, job satisfaction refers to the degree to which nurses’ professional expectations and needs align with their work reality [9,10,11]. Key determinants of job satisfaction include workload, resource availability, professional recognition, and organizational support. In hospital settings, these factors have been extensively studied, revealing that nurses often experience significant levels of stress, workload, and emotional demands, which affect their job satisfaction [12,13]. However, it is important to recognize that school nurses operate within a unique context that differs substantially from hospital environments. School nurses are often the only healthcare professionals in a school, which can potentially lead to professional isolation. Their roles focus primarily on public health, health promotion, and preventive care, in contrast to hospital nurses who provide secondary and tertiary care within multidisciplinary teams [14,15]. Therefore, the factors influencing job satisfaction among school nurses may differ from those affecting hospital nurses, necessitating dedicated research to understand their specific challenges and needs. Moreover, the extent of professional interaction among school nurses can significantly influence their job satisfaction. Nurses working in isolation, particularly those employed by community or private organizations, may have fewer opportunities for peer support and collaboration [16]. This lack of interaction can lead to feelings of professional isolation, reducing job satisfaction and potentially impacting the quality of care provided to students.

Professional quality of life is another critical aspect that directly influences nurses’ well-being and the quality of care they provide. This concept encompasses not only job satisfaction but also the physical, emotional, and social well-being of professionals [16,17,18,19]. Maintaining a proper balance between work responsibilities and personal life, along with access to supportive resources and a healthy work environment, is essential for a good quality of life. In hospital settings, it has been shown that quality of life is closely linked to job satisfaction and, ultimately, to the quality of care nurses can provide to their patients [20,21]. In the school context, although fewer studies exist, it is reasonable to assume that factors such as work demands, contractual conditions, and institutional support, as highlighted by Cabezas Peña (2000), also affect the quality of life of school nurses. These professionals face challenges such as limited resources, high workloads, and variable collaboration with educational staff, which impact both their well-being and their ability to provide effective care in environments with increasingly complex student health needs [22]. Despite the relevance of these issues, there is a notable gap in research on job satisfaction and quality of life among school nurses in Spain. While extensively explored in other contexts, such as hospitals, the school setting has received less attention in the scientific literature. This gap is concerning, as the working conditions and well-being of school nurses affect not only their performance but also the quality of care that students receive. Therefore, conducting studies that analyze the factors influencing job satisfaction and quality of life among school nurses in Spain is important, as it may help identify areas for improvement with regard to their working conditions, which could potentially have a positive impact on the educational healthcare environment [23,24].

The role of the school nurse is unique and multifaceted, encompassing direct student care, health promotion, and health education. School nurses must adapt to diverse educational environments, collaborate with educational professionals, and often manage limited resources [25,26,27,28]. This variety of responsibilities and the context in which they operate can significantly influence their job satisfaction. Factors such as autonomy in decision-making, professional recognition within the educational setting, opportunities for professional development, and the quality of interpersonal relationships with teaching and administrative staff are key determinants in the perception of job satisfaction among school nurses [28,29,30,31,32]. Understanding how these specific aspects of the role affect their professional well-being is essential for improving not only their quality of work life but also the quality of care provided to students [28,33].

This study aims to evaluate the perception of job satisfaction and professional quality of life among school nurses in Spain. Our specific contributions include addressing a significant gap in the existing literature, which has predominantly focused on hospital settings, thereby overlooking the unique challenges faced by school nurses. Additionally, we employed a robust quantitative approach with a large sample size, enhancing the representativeness and generalizability of our findings.

## 2. Materials and Methods

### 2.1. Study Design

A cross-sectional descriptive study with a quantitative approach was conducted in Spain between November 2023 and February 2024. This design was chosen to provide an overview of the job satisfaction and quality of working life of school nurses at a specific point in time. The cross-sectional approach allows for identifying patterns and associations within a diverse sample, facilitating comparisons with future studies. This design was selected for its efficiency in collecting data from a large sample within a short timeframe.

### 2.2. Study Population

In Spain, school health services are a collaborative effort involving nurses working directly within schools and those from community and public health sectors. For this study, all nurses providing health services to students, regardless of their primary workplace, were classified as “school nurses”.

The study population included school nurses from various geographical regions and educational settings across Spain, as defined by the National School Nursing Observatory of the General Nursing Council of Spain [25]. This population encompassed nurses directly employed by educational institutions (primary, secondary, and special education schools) and those working in community/public health settings or private companies that provide school health services. By including nurses from these diverse settings, the study aimed to capture the full spectrum of professionals involved in school healthcare in Spain, reflecting the country’s integrated approach to student health services.

### 2.3. Inclusion Criteria

Nurses included in the study were those providing health services to educational centers, regardless of their primary workplace. This group consisted of the following:
Nurses directly employed by educational institutions, such as primary, secondary, and special education schools, who work on-site as part of the school team.Nurses contracted by external entities, such as community organizations, public health services, or private companies, who regularly deliver healthcare services to students in schools under formal agreements.

Additionally, all participants were required to have at least one year of professional experience in this field to ensure sufficient adaptation and familiarity with their work environment. This criterion was designed to ensure that responses reflected a consolidated perception of their job satisfaction and quality of working life.

### 2.4. Sample Size and Participants

A power calculation was conducted to determine the required sample size for detecting significant differences in job satisfaction and professional quality of life, with a 95% confidence level and a 5% margin of error. The initial calculation indicated a requirement of 328 participants. Ultimately, 553 school nurses were recruited using convenience sampling to enhance the robustness of the study and improve the external validity of the results.

Informed consent was obtained from all participants, adhering to the principles of the Declaration of Helsinki and the Belmont Report. The study was approved by the Professional Ethics Committee of the University of Vic-UCC (code 303/2023). Participation was entirely voluntary, and anonymity and data confidentiality were ensured.

The increased sample size aimed to minimize potential biases, address heterogeneity within the target population, and improve the precision of confidence intervals, thereby reinforcing the reliability and representativeness of the findings.

### 2.5. Ethical Procedures

Participants were fully informed about the study’s objectives and procedures, as well as their right to withdraw at any time without repercussions. Informed consent was obtained prior to participation, emphasizing the voluntary and anonymous nature of their involvement. Data were used exclusively for research purposes, and measures were implemented to minimize any inconvenience or time burden for participants. From an ethical perspective, recruiting a larger sample size was justified to achieve more precise and generalizable results.

### 2.6. Instruments

Font-Roja job satisfaction questionnaire: This validated instrument measures job satisfaction using 26 items across nine dimensions as follows: job satisfaction, work-related tension, professional competence, work pressure, opportunities for improvement, interpersonal relationships with supervisors, interpersonal relationships with colleagues, extrinsic status characteristics, and job monotony. Responses are rated on a Likert scale ranging from 1 (minimal satisfaction) to 5 (maximum satisfaction). It has been validated in nursing contexts with a Cronbach’s alpha coefficient exceeding 0.80 [26].

CPV-35 professional quality of life questionnaire: Designed for healthcare professionals, the CPV-35 assesses professional quality of life using a 7-point Likert scale. It evaluates dimensions such as psychological well-being, social support at work, satisfaction with the work environment, and work–life balance. Key aspects include supervisor support, workload, salary satisfaction, recognition, interpersonal relationships, autonomy, and opportunities for professional development. It has been validated with a Cronbach’s alpha coefficient exceeding 0.85 [10].

### 2.7. Data Collection Procedure

Questionnaire design: In addition to the Font-Roja and CPV-35 instruments, demographic and occupational questions (e.g., age, gender, years of experience, and weekly working hours) were included.

Administration: Surveys were distributed electronically via links shared through school nursing groups and associations, ensuring confidentiality and anonymity of responses.

Quality control: Procedures included the pre-distribution validation of the questionnaires and reviewing the data collected to address missing or inconsistent responses. Data collection took place between November 2023 and February 2024.

### 2.8. Statistical Analysis

Descriptive statistics: Sample characteristics were described using absolute and relative frequencies for categorical variables and medians with interquartile ranges (IQR) for numerical variables. Font-Roja and CPV-35 scores were presented in tables using the medians [IQR] and means with standard deviations (SD).

Comparative analysis: Scores were compared based on contracting entity and years of experience using chi-square tests for categorical variables, Kruskal–Wallis tests for the medians [IQR], and ANOVA for the means (SD).

Regression models: Multivariate regression models were developed to explain the total scores of the Font-Roja and CPV-35 questionnaires using variables such as gender, age, experience, employment status, contracting entity, inter-institutional relationships, and weekly working hours. Backward stepwise variable selection based on the Akaike Information Criterion was applied to obtain reduced models with optimal explanatory power.

Nonlinear associations: Associations between CPV-35 factors and the total Font-Roja score were analyzed using spline regression models, with *p*-values calculated via likelihood ratio tests. Similar analyses explored the relationship between Font-Roja factors and the CPV-35 score.

#### Categories of Contracting Entities

Participants were classified into four contracting categories, considering all the entities that employ school nurses in Spain, as follows: ’Global’, ‘Health’, ‘Education’, and ‘Other This classification enabled comparisons across different contexts as follows: ‘Global’ included all participants, providing an overview of the results. ‘Health’ encompassed nurses employed by health sector entities, such as hospitals and public health centers. ‘Education’ included nurses hired by public and private educational institutions. ‘Other’ referred to nurses employed by private companies providing school nursing services.

### 2.9. Theoretical Framework

This study integrates personal attributes, job characteristics, and organizational factors into its theoretical framework as follows: Personal attributes: Demographic and professional variables (e.g., age, gender, years of experience). Job characteristics: Intrinsic aspects of the nursing role, such as workload, autonomy, professional development opportunities, and interpersonal relationships. Organizational factors: Broader employment contexts, including employment status, contracting entity, and inter-institutional collaboration.

This framework provided a comprehensive perspective for examining how individual and organizational factors influence job satisfaction and professional quality of life. It guided variable selection, hypothesis formulation, and result interpretation, offering a nuanced understanding of the dynamics of school nursing in Spain.

## 3. Results

The study included 553 school nurses, with 63.3% employed by the Education sector, 12.1% by the health sector, and 24.6% by other entities. As shown in Table 1, the majority of participants were female (97.5%), with no significant differences across employing entities (*p* = 0.474). The average age of participants was consistent across sectors, approximately 40 years, with no statistically significant differences (*p* = 0.824).

Regarding work experience, the mean was 4 years, and most nurses had less than 5 years of experience (91.9%), with no significant differences among the entities (*p* = 0.402). Although a trend in experience distribution was noted, it was not statistically significant (*p* = 0.087).

Employment status differed significantly by employing entity (*p* < 0.001). In the Health sector, 25.4% were permanent employees, as compared to 8.0% in Education and 58.8% in other entities. Temporary employment was most common in Health (49.3%) and nearly absent in other entities.

There were also significant differences in the hiring authority depending on the employing entity (*p* < 0.001). In the Health sector, 89.6% were employed by the Ministry of Health, while 41.4% in Education were employed by the Ministry of Education, and 46.3% in other entities were employed directly by their educational institution.

Collaboration with other institutions showed significant variation (*p* = 0.023). In Health, 78.6% collaborated with primary care, compared to 58.0% in Education and 51.7% in other entities. Additionally, 48.3% of those in other entities reported relationships with various institutions. Weekly working hours were consistent across entities, averaging 37.5 h, with no significant differences (*p* = 0.362).

Table 2 presents the descriptive characteristics of the sample centers, broken down by contracting entity. The analysis shows significant variation in the number of students across different entities (*p* = 0.037). The majority of schools (59.8%) have between 501 and 1500 students. This distribution does not show significant differences among the Health, Education, and other entities, reflecting the wide range of school sizes within the study sample. However, schools with more than 1500 students are more common in the “Others” category (22.1%). The type of school shows significant differences among contracting entities (*p* < 0.001). Schools focused on early childhood and primary education are predominantly found in the Education sector (42.6%). In contrast, schools offering early childhood, primary, and secondary education are more common in the “Others” category (55.9%). This indicates that the “Others” category includes a broader range of educational levels. Ownership of the school varies significantly by entity (*p* = 0.003). Public schools are the most common in the Education sector (76.6%), whereas privately funded schools are more prevalent in the “Others” category (22.1%).

Regarding the characteristics of the educational institution, there are no significant differences among the entities (*p* = 0.131). The majority of schools across all entities are located in urban areas, with 89.7% of schools being urban and 10.3% rural.

In Table 3, significant differences are observed between contracting entities in two specific factors. Factor 8, which assesses extrinsic status characteristics, shows a statistically significant difference (*p* = 0.025). This suggests that perceptions of job status vary across different entities, with notable distinctions in how job status is perceived. Similarly, Factor 9, which measures perceptions of job monotony, also exhibits a significant difference (*p* = 0.038). These findings indicate that job monotony is perceived differently depending on the contracting entity. In contrast, the other factors analyzed—Factor 1: Job satisfaction, Factor 2: Work-related tension, Factor 3: Professional competence, Factor 4: Work pressure, Factor 5: Improvement opportunities, Factor 6: Interpersonal relationship with superiors, Factor 7: Interpersonal relationship with colleagues, and the overall Font-Roja questionnaire score—do not show statistically significant differences between contracting entities. This consistency suggests a uniform perception regarding job satisfaction, work-related tension, professional competence, work pressure, opportunities for improvement, and interpersonal relationships across the evaluated entities.

Table 4 highlights several important findings regarding the professional quality of life questionnaire (CVP-35) by contracting entity. The perceived workload was significantly higher in the “Others” sector (7.79) compared to Health (7.16) and Education (7.45) (*p* = 0.026). Satisfaction with salary was notably higher in the Education sector (6.55) than in the Health (5.93) and other sectors (5.67) (*p* = 0.020). The pressure to maintain the quantity of work was also higher in Health (5.64) compared to other sectors (4.81) (*p* = 0.044), and similarly, the pressure to maintain quality of work was higher in Health (5.63) compared to other sectors (5.35) (*p* = 0.037). Support from supervisors was greater in the Education sector (6.53) compared to other sectors (7.06) (*p* = 0.032). Additionally, the perception of a lack of time for personal life was higher in the Education sector (4.16) compared to other sectors (3.78) (*p* = 0.022). Receiving information about work results was higher in the “Others” sector (5.19) compared to Health (4.93) and Education (4.39) (*p* = 0.012). The importance of work to the lives of others was notably higher in the Education sector (9.15) compared to Health (8.67) and other sectors (8.68) (*p* = 0.003). Finally, the necessity for training was greater in the “Others” sector (7.89) compared to Health (7.36) and Education (7.91) (*p* = 0.032).

The multivariate regression analysis, as shown in Table 5, on the total scores of the Font-Roja and CVP-35 questionnaires revealed significant associations, providing relevant insights into job satisfaction and professional quality of life.

Font-Roja questionnaire analysis: The analysis indicated that age and weekly working hours are significantly associated with the total score of the Font-Roja questionnaire. Specifically, individuals aged between 30 and 40 years showed a decrease in the total questionnaire score, with a *p*-value of 0.045, indicating an average reduction of 0.084 points (95% CI: −0.166, −0.002). This suggests that in this age group, job satisfaction and professional quality of life may be negatively affected.

Furthermore, those working more than 37.5 h per week also showed a decrease in the total score, with a *p*-value of 0.076, which is close to the conventional significance threshold of 0.05. The observed reduction was 17.1% compared to those working less than 15 h per week (95% CI: −0.359, 0.018). This negative association suggests that working many hours could be related to lower job satisfaction and professional quality of life.

It is important to note that, although the *p*-value for weekly working hours (0.076) does not reach the traditional level of statistical significance, it is close enough to suggest a possible trend that might be significant with a larger sample size or in other contexts. In practical terms, both age and working hours seem to negatively influence the Font-Roja questionnaire scores, highlighting the need to consider these factors in managing workplace well-being.

CVP-35 questionnaire analysis: In the analysis of the CVP-35, it was found that the relationship with institutions such as the city council was negatively associated with the total score, with this association being highly significant (*p* = 0.002). This suggests that individuals whose employment relationship is related with these public institutions tend to have a more negative perception of their job satisfaction and professional quality of life.

Additionally, the “non-binary” gender category showed a significant negative association with the total CVP-35 score (*p* = 0.048). Specifically, non-binary individuals demonstrated a decrease of 1.616 points in the total questionnaire score (95% CI: −3.218, −0.013). This finding indicates that non-binary individuals may experience lower job satisfaction and professional quality of life compared to their male and female counterparts.

These associations underscore the importance of considering institutional relationships and gender identity when evaluating job satisfaction and professional quality of life. Specifically, the policies and practices of the city council, as well as the recognition and support of non-binary individuals, can have a significant impact on these aspects.

It was also found that individuals working between 15 and 30 h per week showed a significant positive association with the total score in the reduced model (*p* = 0.035), with an increase of 0.537 points compared to those working less than 15 h per week (95% CI: 0.037, 1.037). Similarly, those working more than 37.5 h per week showed a significant positive association (*p* = 0.011), with an increase of 0.591 points compared to those working less than 15 h per week (95% CI: 0.134, 1.048).

Model details: The reduced model achieves the best balance of likelihood by the number of variables, as a result of a stepwise backward variable selection process based on the Akaike Information Criterion (AIC). Although the *p*-values of the marginal effects may not always present a *p* < 0.05, the model without those variables had a worse AIC. No reduced model is presented for Font-Roja since none of the explanatory variables provided sufficient likelihood when trying to explain the total Font-Roja score.

In Figure 1, it can be observed that Factor 5, “Improvement options”, from the Font-Roja job satisfaction questionnaire is the only one that shows a statistically significant association with the total score of the professional quality of life questionnaire CVP-35, with a *p*-value of 0.016. Other factors, such as Factor 1, “Job satisfaction”, also approach statistical significance with a *p*-value of 0.063. When applying a reduced multivariate regression model, which includes the nonlinear relationship detected in Factor 5 and other factors with linear effects, significant associations are identified with Factors 1 and 9, with *p*-values of 0.046 and 0.008, respectively. In this model, Factor 1 has a proportional association and Factor 9 an inverse association with professional quality of life. The model that best explains the total CVP-35 score, according to the Akaike Information Criterion (AIC), includes Factors 1, 2, 5, 8, and 9. Although Factors 2 and 8 do not reach statistical significance due to their *p*-values (>0.05), their inclusion in the model remains relevant, with Factor 5 showing a predominant nonlinear relationship.

## 4. Discussion

The average score of some factors related to job satisfaction and professional quality of life for school nurses shows significant differences based on their employing entity.

The findings indicate that, overall, school nurses have a mean score of 3.129, consistent with previous studies reporting medium-high levels of satisfaction [27]. However, significant variations are observed in specific factors, such as perceived status and job monotony, across different sectors. Nurses in the Health sector perceive having a higher job status, likely due to the formal recognition and structure of the healthcare system. In contrast, those in the Education sector experience greater job monotony, potentially influenced by routine tasks and limited variety in the school environment [8,28].

Work-related stress is a common experience among school nurses across all sectors, reflecting the inherently demanding nature of the profession. In our study, this aspect was assessed using Factor 2: Work-related tension from the Font-Roja job satisfaction questionnaire, which yielded a mean score of 2.45 (Table 3). This result indicates moderate levels of tension among the nurses, aligning with the existing literature that highlights the stress inherent in school nursing [29,30]. The perception of professional competence is homogeneous nationwide, suggesting that the training and skills acquired are adequate. Professional competence is crucial for job satisfaction and overall well-being [31].

The pressure to maintain both the quantity and quality of work is a consistent concern, with a mean score of 2.97 in Factor 4: Work pressure (Table 3). This suggests that, despite the relatively high job satisfaction (mean score of 3.85 in Factor 1: Job satisfaction), nurses face considerable demands that can affect their satisfaction and professional quality of life. The “Others” sector shows a more positive perception regarding improvement opportunities, possibly due to greater flexibility compared to the Health and Education sectors. This highlights the need for developing and communicating development opportunities across all sectors to enhance satisfaction and professional quality of life [32].

Interpersonal relationships with supervisors and colleagues are generally positive and similar across sectors, suggesting a uniform collaborative culture. However, the greater job monotony in the Education sector underscores the need to diversify roles to reduce monotony and improve job satisfaction [34].

Moreover, our findings suggest that nurses who are less integrated within the educational sphere and have limited interaction with other school nurses may experience lower job satisfaction. This could be due to a lack of involvement in educational activities, reduced opportunities for collaboration with educational staff, and fewer chances to engage with professional peers. These factors may contribute to feelings of marginalization and professional isolation, which are known to negatively affect job satisfaction [33].

Our findings underscore the unique challenges faced by school nurses. The professional isolation inherent in being the sole healthcare provider in a school can contribute to feelings of marginalization and impact job satisfaction [35]. Additionally, the focus on public health and preventive care requires a different set of skills and may not receive the same recognition as acute care provided in hospitals. These factors highlight the need for specialized support and resources tailored to the school nursing context [25,28,35].

Professional quality of life varies by sector. The perceived workload is higher in the “Others” sector, potentially due to the diversity of tasks and less regulation compared to more structured sectors. Job satisfaction with salary is higher in the Education sector, which may reflect better salary conditions or benefits. The pressure to maintain the quantity and quality of work is significantly higher in the Health sector, reflecting the demanding nature of the healthcare environment. Higher supervisor support in the Education sector suggests a more collaborative and less hierarchical work environment [33,34].

Collaboration with primary care is more frequent in the Health sector, aligning with the integration into the healthcare system. In contrast, collaboration in the Education and “Others” sectors is less frequent, which may limit the support and resources available to school nurses in these settings. Additionally, the presence of larger educational centers in the “Others” sector affects the workload and work dynamics of school nurses [33].

Nurses with 5 to 10 years of experience report a lower professional quality of life, possibly due to unmet expectations and job realities that generate dissatisfaction. This finding is consistent with previous studies suggesting that less experienced nurses are generally more satisfied [34].

It is important to highlight that less experienced nurses often have limited opportunities to express dissatisfaction, a factor closely tied to the type of employment contract. Frequently employed under more precarious contracts, less experienced nurses are more likely to work longer hours and be assigned more complex tasks. This dynamic is further exacerbated by excessive workloads, which, while positively associated with the overall professional quality of life, show a stronger negative impact on job satisfaction. These findings emphasize the need to address workload issues and promote work–life balance as critical measures to enhance job satisfaction [7].

In the school context, although research is more limited, it is reasonable to assume that similar factors influence the quality of life of school nurses. As highlighted by Cabezas Peña (2000), work demands, contractual conditions, and institutional support significantly impact professional well-being. School nurses often face specific challenges, such as limited resources, high workloads, and variable collaboration with educational staff. These factors not only affect their well-being but also hinder their ability to provide effective care, particularly in environments with increasingly complex student health needs [22]. Additionally, age significantly impacts job satisfaction. Individuals aged 30 to 40 exhibit lower satisfaction levels, which may suggest that this age group faces unique challenges within their work environment. Targeted interventions could address these specific challenges, improving satisfaction and overall professional quality of life [35].

This study aims to evaluate the perception of job satisfaction and professional quality of life among school nurses in Spain, addressing a significant gap in the existing literature, which has predominantly focused on hospital settings. By examining the unique challenges faced by school nurses—such as work demands, contractual conditions, and institutional support—and employing a robust quantitative approach with a large sample size, this research provides a comprehensive understanding of the factors influencing their professional experiences. These contributions enhance the representativeness and generalizability of the findings, offering valuable insights for improving the well-being and effectiveness of school nurses.

### 4.1. Practical Implications and Recommendations

The study underscores the importance of tailoring support policies and programs to the specific needs of each sector. For instance, in the Education sector, diversifying responsibilities and formally recognizing nurses’ contributions could address job monotony and improve status perception. Furthermore, strengthening collaboration between the Education and Health sectors could enhance support and resources available to school nurses, while implementing professional development programs and promotion opportunities could improve motivation and reduce monotony.

### 4.2. Limitations of the Study

This study has several limitations. The cross-sectional design prevents establishing causal relationships between variables, limiting conclusions about the direct influence of certain factors on job satisfaction. Additionally, while the sample size was robust, data were only collected from Spain, which may restrict the generalizability of the findings to other international contexts. The self-selection of participants and the use of self-reported questionnaires introduce risks of bias, such as social desirability and the potential underrepresentation of more burdened nurses. Finally, the lack of a longitudinal follow-up restricts the ability to observe how job satisfaction and professional quality of life evolve over time.

### 4.3. Future Research Directions and Implications

Future research should explore contextual factors affecting school nurses’ job satisfaction, design and implement targeted interventions to enhance their work well-being, and examine how job satisfaction impacts the quality of school health services. Comparative studies across countries, longitudinal research, and analyses on how job satisfaction influences service quality could provide a more comprehensive understanding and contribute to international improvements in school healthcare. Additionally, qualitative research is needed to gain deeper insights into the specific experiences and needs of these health professionals, as well as to identify barriers and facilitators of job satisfaction. Combining qualitative and quantitative methods should offer a more holistic and nuanced view, facilitating the development of more effective strategies to support school nurses.

## 5. Conclusions

School nurses in Spain generally report a moderately high level of job satisfaction, although this varies across employment sectors—Health, Education, and Others. These differences are linked to specific working conditions and challenges within each sector. While the overall professional quality of life, measured by the CPV-35 scale, averages 6.000, common work-related pressures are observed across all sectors.

Nurses in the Health sector perceive having higher job status and facing greater pressure to maintain work quantity and quality, reflecting the demanding nature of the healthcare environment. In contrast, the Education sector shows greater job monotony, affecting job satisfaction, though this is somewhat mitigated by higher perceived support from supervisors. The “Others” sector reports a higher perceived workload but also more opportunities for improvement, possibly due to a less regulated work environment.

Experienced nurses place significant value on recognition and professional competitiveness. Working more than 37.5 h per week is associated with a better professional quality of life, although it may negatively impact job satisfaction due to potential overwork.

The variability in the presence of school nurses across different models, where only about half of schools have a dedicated nurse, highlights the need for policies that ensure the equitable distribution of nursing services. Additionally, providing health training to teaching staff and strengthening collaboration between the educational and healthcare sectors could enhance support for nurses and student health.

To improve the well-being of school nurses, it is crucial to enhance their integration within the educational system and foster professional networks. Developing programs that promote collaboration with educational staff and facilitate interaction among school nurses can address feelings of isolation and improve job satisfaction. Future research should focus on exploring these contextual factors and designing interventions that enhance both occupational well-being and the quality of school health services. This will contribute to a more comprehensive understanding of school nursing dynamics and support the development of effective policies and practices.

## Figures and Tables

**Figure 1 healthcare-13-00323-f001:**
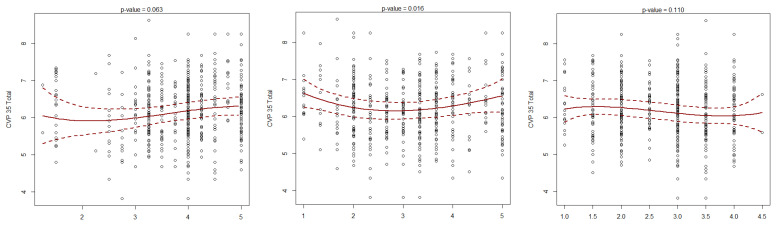
Association between the factors of the Font-Roja job satisfaction questionnaire and the total score of the professional quality of life questionnaire CVP-35. Factor 1: Job satisfaction; Factor 5: Improvement options; Factor 9: Job monotony.

**Table 1 healthcare-13-00323-t001:** Characteristics of participants according to employing entity.

Characteristic	Global (*n* = 553)	Health (*n* = 67)	Education (*n* = 350)	Others * (*n* = 136)	*p*-Value
Gender (*n* = 553)					0.474
Male	13 (2.4%)	3 (4.5%)	9 (2.6%)	1 (0.7%)	
Female	539 (97.5%)	64 (95.5%)	340 (97.1%)	135 (99.3%)	
Non-binary	1 (0.2%)	0 (0.0%)	1 (0.3%)	0 (0.0%)	
Age (*n* = 553)	40.0 [33.0; 46.0]	40.0 [34.0; 45.0]	41.0 [33.3; 45.2]	40.0 [33.0; 46.0]	0.824
Experience (years) (*n* = 553)	4.0 [2.0; 7.0]	4.0 [2.0; 7.5]	4.0 [2.0; 6.0]	4.0 [3.0; 8.0]	0.402
Experience (years, categorized) (*n* = 553)					0.087
<5 years	508 (91.9%)	63 (94.0%)	322 (92.0%)	123 (90.4%)	
5 to 10 years	30 (5.4%)	1 (1.5%)	20 (5.7%)	9 (6.6%)	
10 to 15 years	7 (1.3%)	0 (0.0%)	4 (1.1%)	3 (2.2%)	
15 to 20 years	6 (1.1%)	3 (4.5%)	3 (0.9%)	0 (0.0%)	
>20 years	2 (0.4%)	0 (0.0%)	1 (0.3%)	1 (0.7%)	
Employment status (*n* = 553)					<0.001
Discontinuous **	266 (48.1%)	17 (25.4%)	204 (58.3%)	45 (33.1%)	
Permanent	125 (22.6%)	17 (25.4%)	28 (8.0%)	80 (58.8%)	
Temporary	130 (23.5%)	33 (49.3%)	97 (27.7%)	0 (0.0%)	
Non-permanent/Occasional	32 (5.8%)	0 (0.0%)	21 (6.0%)	11 (8.1%)	
Hiring authority (*n* = 553)					<0.001
City council	7 (1.3%)	0 (0.0%)	0 (0.0%)	7 (5.1%)	
Education Department	145 (26.2%)	0 (0.0%)	145 (41.4%)	0 (0.0%)	
Health Department	60 (10.8%)	60 (89.6%)	0 (0.0%)	0 (0.0%)	
Education	205 (37.1%)	0 (0.0%)	205 (58.6%)	0 (0.0%)	
School nursing company	31 (5.6%)	0 (0.0%)	0 (0.0%)	31 (22.8%)	
Others	35 (6.3%)	0 (0.0%)	0 (0.0%)	35 (25.7%)	
Own educational center	63 (11.4%)	0 (0.0%)	0 (0.0%)	63 (46.3%)	
Health	7 (1.3%)	7 (10.4%)	0 (0.0%)	0 (0.0%)	
Relationship with other institutions (*n* = 124)					0.023
Primary care	73 (58.9%)	11 (78.6%)	47 (58.0%)	15 (51.7%)	
City council	11 (8.9%)	0 (0.0%)	11 (13.6%)	0 (0.0%)	
Others	38 (30.6%)	2 (14.3%)	22 (27.2%)	14 (48.3%)	
Social services	2 (1.6%)	1 (7.1%)	1 (1.2%)	0 (0.0%)	
Weekly hours (*n* = 227)	37.5 [35.0; 37.5]	37.5 [30.0; 37.5]	37.5 [35.0; 37.5]	37.0 [33.8; 38.5]	0.362

* Others: Private health companies in the healthcare sector, associations, etc. ** Discontinuous work involves employment with irregular or intermittent hours. Statistical analysis was performed using Chi-square tests for categorical variables and ANOVA for continuous variables.

**Table 2 healthcare-13-00323-t002:** Descriptive characteristics of sample centers—globally and by contracting entity.

	Global (*n* = 553)	Health (*n* = 67)	Education (*n* = 350)	Others * (*n* = 136)	*p*-Value
Number of students in the center (*n* = 552)					0.037
≤500	144 (26.1%)	19 (28.8%)	91 (26.0%)	34 (25.0%)	
501 to 1500	330 (59.8%)	38 (57.6%)	220 (62.9%)	72 (52.9%)	
>1500	78 (14.1%)	9 (13.6%)	39 (11.1%)	30 (22.1%)	
Type of center					<0.001
High school	6 (1.1%)	0 (0.0%)	5 (1.4%)	1 (0.7%)	
Special education center	45 (8.1%)	14 (20.9%)	25 (7.1%)	6 (4.4%)	
Early childhood and primary	205 (37.1%)	20 (29.9%)	149 (42.6%)	36 (26.5%)	
Early childhood, primary, and secondary	209 (37.8%)	20 (29.9%)	113 (32.3%)	76 (55.9%)	
Primary education	32 (5.8%)	6 (9.0%)	19 (5.4%)	7 (5.1%)	
Nursery school	4 (0.7%)	1 (1.5%)	3 (0.9%)	0 (0.0%)	
Regular school with special education	4 (0.7%)	0 (0.0%)	2 (0.6%)	2 (1.5%)	
Secondary education institute	48 (8.7%	6 (9.0%)	34 (9.7%)	8 (5.9%)	
Ownership of the center					0.003
State-subsidized private school	80 (14.5%)	8 (11.9%)	42 (12.0%)	30 (22.1%)	
Private school	69 (12.5%)	5 (7.5%)	40 (11.4%)	24 (17.6%)	
Public school	404 (73.1%)	54 (80.6%)	268 (76.6%)	82 (60.3%)	
Center characteristics					0.131
Rural	57 (10.3%)	9 (13.4%)	40 (11.4%)	8 (5.9%)	
Urban	496 (89.7%)	58 (86.6%)	310 (88.6%)	128 (94.1%)	

Others *: Private health companies in the healthcare sector, associations, etc. Statistical analysis was conducted using descriptive statistics (mean and standard deviation) and ANOVA to compare differences between contracting entities.

**Table 3 healthcare-13-00323-t003:** Average Font-Roja job satisfaction questionnaire scores by contracting entity.

Variables	Global	Health (*n* = 67)	Education (*n* = 350)	Others * (*n* = 136)	*p*-Value
Factor 1: Job satisfaction	3.85 (0.83)	3.74 (0.87)	3.85 (0.84)	3.93 (0.80)	0.302
Factor 2: Work-related tension	2.45 (0.51)	2.47 (0.51)	2.45 (0.51)	2.44 (0.50)	0.925
Factor 3: Professional competence	1.55 (0.73)	1.58 (0.65)	1.57 (0.76)	1.49 (0.69)	0.511
Factor 4: Work pressure	2.97 (0.76)	3.13 (0.76)	2.96 (0.76)	2.95 (0.76)	0.200
Factor 5: Improvement opportunities	3.11 (1.05)	3.30 (1.07)	3.04 (1.03)	3.19 (1.07)	0.096
Factor 6: Interpersonal relationship with superiors	3.20 (1.13)	3.28 (1.13)	3.21 (1.11)	3.15 (1.19)	0.748
Factor 7: Interpersonal relationship with colleagues	2.49 (1.12)	2.45 (1.06)	2.49 (1.13)	2.52 (1.14)	0.911
Factor 8: Extrinsic status characteristics	2.78 (0.89)	3.03 (0.84)	2.72 (0.93)	2.82 (0.81)	0.025
Factor 9: Job monotony	2.74 (0.83)	2.59 (0.76)	2.81 (0.82)	2.64 (0.87)	0.038
Total Font-Roja score	2.86 (0.31)	2.91 (0.28)	2.85 (0.32)	2.87 (0.31)	0.39

Others *: Private health companies in the healthcare sector, associations, etc. Statistical analysis was carried out using ANOVA to assess differences in job satisfaction scores among contracting entities.

**Table 4 healthcare-13-00323-t004:** Professional quality of life (CVP-35) questionnaire scores—overall and by contracting entity (mean (SD)).

Variables	Global	Health (*n* = 67)	Education (*n* = 350)	Others * (*n* = 136)	*p*-Value
Amount of work I have	7.50 (1.67)	7.16 (1.76)	7.45 (1.63)	7.79 (1.67)	0.026
Satisfaction with the type of work	7.87 (1.78)	7.82 (1.56)	7.89 (1.79)	7.86 (1.86)	0.960
Satisfaction with salary	5.94 (2.12)	6.55 (2.01)	5.93 (2.08)	5.67 (2.21)	0.020
Opportunity for promotion	2.83 (2.42)	3.33 (2.81)	2.69 (2.34)	2.95 (2.43)	0.117
Recognition of my effort	5.68 (2.51)	5.67 (2.55)	5.57 (2.52)	5.96 (2.47)	0.304
Pressure to maintain quantity of work	4.92 (2.53)	5.64 (2.32)	4.81 (2.52)	4.86 (2.62)	0.044
Pressure to maintain quality of work	5.09 (2.55)	5.63 (2.37)	4.89 (2.53)	5.35 (2.63)	0.037
Hurry and stress due to lack of time for my work	5.55 (2.59)	5.64 (2.54)	5.32 (2.55)	6.10 (2.67)	0.011
Motivation (willingness to exert effort)	7.75 (2.05)	7.78 (1.98)	7.71 (2.04)	7.85 (2.11)	0.807
Support from my supervisors	6.61 (2.58)	6.12 (2.66)	6.53 (2.62)	7.06 (2.37)	0.032
Due to support from my colleagues, I feel I am at the limit in several aspects	4.59 (2.62)	4.63 (2.68)	4.48 (2.60)	4.85 (2.65)	0.378
Support from my family	8.54 (2.15)	8.57 (2.08)	8.60 (2.10)	8.38 (2.30)	0.574
Desire to be creative	8.07 (2.09)	8.27 (1.86)	8.05 (2.11)	8.01 (2.14)	0.695
Opportunity to be creative	6.21 (2.54)	6.36 (2.46)	6.16 (2.56)	6.26 (2.56)	0.809
Disconnecting after work	6.29 (2.72)	6.16 (2.77)	6.37 (2.70)	6.15 (2.77)	0.675
Receiving information about the results of my work	4.65 (2.81)	4.93 (2.85)	4.39 (2.74)	5.19 (2.90)	0.012
Conflicts with other people at work	3.03 (2.39)	3.10 (2.52)	2.98 (2.35)	3.12 (2.43)	0.825
Lack of time for my personal life	3.56 (2.41)	4.16 (2.54)	3.37 (2.34)	3.78 (2.48)	0.022
Physical discomfort at work	3.62 (2.59)	3.75 (2.66)	3.54 (2.50)	3.76 (2.79)	0.639
Opportunity to express what I think and need	6.35 (2.49)	6.16 (2.54)	6.39 (2.55)	6.32 (2.31)	0.785
Responsibility load	8.41 (1.82)	8.09 (2.27)	8.54 (1.64)	8.24 (2.00)	0.085
My organization tries to improve the quality of life in my position	4.51 (2.74)	4.37 (2.42)	4.40 (2.78)	4.88 (2.79)	0.196
I have autonomy or freedom of decision	6.86 (2.31)	6.84 (2.46)	6.81 (2.31)	7.01 (2.23)	0.680
Annoying interruptions	5.55 (2.71)	5.13 (2.72)	5.67 (2.66)	5.44 (2.84)	0.284
Stress (emotional effort)	6.08 (2.55)	6.06 (2.55)	5.93 (2.57)	6.46 (2.46)	0.121
Necessary training to do my job	7.89 (1.90)	7.36 (2.09)	7.91 (1.84)	8.10 (1.94)	0.032
I am capable of doing my current job	8.71 (1.40)	8.34 (1.57)	8.73 (1.39)	8.85 (1.32)	0.051
Variety in my work	6.68 (2.34)	6.22 (2.55)	6.64 (2.28)	6.99 (2.34)	0.078
My work is important to the lives of others	8.98 (1.60)	8.67 (2.27)	9.15 (1.28)	8.68 (1.86)	0.003
It is possible that my responses are heard and applied	6.35 (2.50)	6.04 (2.57)	6.40 (2.49)	6.39 (2.52)	0.559
What I have to do is clear	6.75 (2.49)	6.99 (2.32)	6.71 (2.46)	6.75 (2.65)	0.712
I am proud of my work	8.67 (1.58)	8.39 (1.49)	8.70 (1.57)	8.74 (1.62)	0.283
My work has negative consequences for my health	3.72 (2.55)	3.93 (2.48)	3.64 (2.57)	3.82 (2.54)	0.616

Others *: Private health companies in the healthcare sector, associations, etc. Statistical analysis was performed using ANOVA to evaluate variations in professional quality of life scores across contracting entities.

**Table 5 healthcare-13-00323-t005:** Multivariate regression models on total scores of Font-Roja and CVP-35 questionnaires.

		Total Font-Roja Score		Total CVP-35 Score		Total CVP-35 Score	
						Reduced Model	
		Marginal Effect (IC95%)	*p*-Value	Marginal Effect (IC95%)	*p*-Value	Marginal Effect (IC95%)	*p*-Value
Independent term		3.129 (2.819, 3.440)	<0.001	6.000 (5.236, 6.763)	<0.001	5.781 (5.201, 6.362)	<0.001
Gender	Male	-ref.-		-ref.-		-ref.-	
	Female	−0.104 (−0.281, 0.072)	0.247	0.138 (−0.296, 0.573)	0.532	0.166 (−0.263, 0.596)	0.447
	Non-binary	0.150 (−0.501, 0.802)	0.651	−1.616 (−3.218, −0.013)	0.048	−1.601 (−3.192, −0.010)	0.049
Age	≤30 years	-ref.-		-ref.-		-ref.-	
	(30, 40] years	−0.084 (−0.166, −0.002)	0.045	0.013 (−0.189, 0.215)	0.900	0.010 (−0.191, 0.211)	0.921
	(40, 50] years	−0.042 (−0.125, 0.040)	0.314	0.103 (−0.100, 0.306)	0.318	0.086 (−0.115, 0.287)	0.403
	>50 years	−0.070 (−0.179, 0.039)	0.207	−0.260 (−0.529, 0.008)	0.057	−0.217 (−0.479, 0.045)	0.104
Experience	<5 years	-ref.-		-ref.-			
	5 to 10 years	0.057 (−0.064, 0.178)	0.356	−0.090 (−0.387, 0.208)	0.554		
	10 to 15 years	−0.076 (−0.318, 0.165)	0.535	0.104 (−0.490, 0.698)	0.731		
	15 to 20 years	0.093 (−0.175, 0.360)	0.497	−0.249 (−0.907, 0.409)	0.458		
	>20 years	0.083 (−0.363, 0.530)	0.714	0.391 (−0.707, 1.490)	0.485		
Employment status	Discontinuous	-ref.-		-ref.-			
	Permanent	0.035 (−0.043, 0.113)	0.377	0.125 (−0.066, 0.317)	0.199		
	Interim	0.025 (−0.044, 0.094)	0.481	−0.053 (−0.224, 0.117)	0.540		
	Non-permanent/temporary	−0.032 (−0.149, 0.085)	0.590	−0.115 (−0.403, 0.173)	0.433		
Hiring authority	Healthcare	-ref.-		-ref.-			
	Education	−0.045 (−0.132, 0.042)	0.309	−0.021 (−0.235, 0.193)	0.847		
	Others	−0.030 (−0.133, 0.073)	0.564	−0.002 (−0.255, 0.251)	0.990		
Relationship with other institutions	Primary care	-ref.-		-ref.-		-ref.-	
	City council	0.190 (−0.013, 0.393)	0.067	−0.801 (−1.301, −0.302)	0.002	−0.825 (−1.316, −0.333)	0.001
	Other	0.026 (−0.099, 0.151)	0.683	−0.157 (−0.465, 0.152)	0.319	−0.184 (−0.483, 0.114)	0.226
	No response	0.072 (−0.023, 0.166)	0.136	0.032 (−0.199, 0.264)	0.783	0.042 (−0.184, 0.269)	0.714
Own records		−0.022 (−0.206, 0.163)	0.817	−0.147 (−0.601, 0.306)	0.524		
Weekly hours	≤15 h	-ref.-		-ref.-		-ref.-	
	(15, 30] h	−0.102 (−0.309, 0.105)	0.333	0.453 (−0.057, 0.963)	0.082	0.537 (0.037, 1.037)	0.035
	(30, 37.5] h	−0.146 (−0.314, 0.022)	0.089	0.234 (−0.180, 0.647)	0.268	0.268 (−0.142, 0.677)	0.199
	>37.5 h	−0.171 (−0.359, 0.018)	0.076	0.538 (0.074, 1.003)	0.023	0.591 (0.134, 1.048)	0.011
	No response	−0.129 (−0.296, 0.039)	0.132	0.168 (−0.244, 0.580)	0.423	0.200 (−0.209, 0.608)	0.338

The reduced model achieves the best balance of likelihood by the number of variables as a result of a stepwise backward variable selection process based on the Akaike Information Criterion (AIC). Although the *p*-values of the marginal effects may not present a *p* < 0.05 in some cases, the model without that variable had a worse AIC. No reduced model is presented for Font-Roja since none of the explanatory variables provided sufficient likelihood when trying to explain the total Font-Roja score.

## Data Availability

The data presented in this study are available upon request from the corresponding author.

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
