# Peer review of "Working Conditions and Well-Being of School Nurses in Spain: Impact on Job Satisfaction and Professional Quality of Life"

_healthcare, 2025, doi:10.3390/healthcare13030323_

Round 1
Reviewer 1 Report
Comments and Suggestions for Authors
Dear Authors,
Congratulations on your work. The article is very well-written, and its organization makes it easy to read. One point worth highlighting is the grounding and relevance of the research (in the introduction section).
My suggestions for revision are specific and relate to the following aspects:
- I encourage the authors to carefully review the bibliographic references. For example, most references are marked with square brackets, but there are also cases with round brackets (see pages 2 and 12).
- On page 5, in the section “Category of contracting entities,” my question is related to understanding why this category was defined a priori. For instance, did the authors consider conducting a cluster analysis to identify "natural groups" within the dataset?
- In the collection of sociodemographic data, did the authors consider gathering information about union membership? This seems important, given research in the field of work psychology that identifies the role of unions as a source of protection for workers' well-being and professional/job satisfaction.
- On page 12, we read that "This suggests that, despite relatively high job satisfaction (mean score of 3.85 in Factor 1: Job Satisfaction), nurses face considerable demands that can affect their satisfaction and professional quality of life”. This sentence well summarizes what work psychology and psychodynamics have long identified: work can be a source of pleasure, satisfaction and recognition and at the same time a source of suffering (due to the demands of the job). If you wish to explore this point further, I recommend to explore the works of Christophe Dejours, for example.
- Still on page 12, the authors mentioned that "Nurses with 5 to 10 years of experience report lower professional quality of life, possibly due to unmet expectations and job realities that generate dissatisfaction. This finding is consistent with previous studies suggesting that less experienced nurses are generally more satisfied." Right, I do not dispute the validity of this statement, but what remains unsaid is that we also know that less experienced nurses have less room to express their dissatisfaction. In this regard, employment status is not irrelevant, of course, as we know that less experienced workers (often with more precarious contracts) tend to work longer hours and are frequently assigned the more difficult and/or more dangerous tasks. This reflection also serves to point out that this is a limitation of purely quantitative research, i.e., apparently, younger nurses have higher satisfaction levels. Fine, great, but do they have the "freedom" (the word may not be the best) to express their dissatisfaction? Perhaps qualitative research could help to understand this issue better.
Once again, congratulations!
Author Response
Revisor 1 Dear Authors,
Congratulations on your work. The article is very well-written, and its organization makes it easy to read. One point worth highlighting is the grounding and relevance of the research (in the introduction section).
Response: We deeply appreciate your comments, and we are sure they improved our article. They are a contribution to our research, and for sure will help to have a higher impact improving the knowledge of school nursing conditions.
My suggestions for revision are specific and relate to the following aspects:
- I encourage the authors to carefully review the bibliographic references. For example, most references are marked with square brackets, but there are also cases with round brackets (see pages 2 and 12).
Response: We have reviewed all the bibliographic references and marked all of them with square brackets as proposed.
- On page 5, in the section “Category of contracting entities,” my question is related to understanding why this category was defined a priori. For instance, did the authors consider conducting a cluster analysis to identify "natural groups" within the dataset?
Response: We have not done a cluster analysis to identify natural groups. We did a previous search to find out who contracts school nurses in Spain and created the categories from that research. Main school nurses are employed by the health system or health entities and from the educational system, and there are few contracted by private companies as it is specified in the article. To make this criterion clearer we added a sentence in page 5 “considering all the entities that employ school nurses in Spain:”
In the collection of sociodemographic data, did the authors consider gathering information about union membership? This seems important, given research in the field of work psychology that identifies the role of unions as a source of protection for workers' well-being and professional/job satisfaction.
Response: Thank you for your comment. Although we recognize the importance of trade unions about workplace well-being, our study did not address this aspect. Considering the article was focused on the availability, perceived need and satisfaction with school nurses, we will consider exploring this relation in future research, especially in qualitative studies.
On page 12, we read that "This suggests that, despite relatively high job satisfaction (mean score of 3.85 in Factor 1: Job Satisfaction), nurses face considerable demands that can affect their satisfaction and professional quality of life”. This sentence well summarizes what work psychology and psychodynamics have long identified: work can be a source of pleasure, satisfaction and recognition and at the same time a source of suffering (due to the demands of the job). If you wish to explore this point further, I recommend to explore the works of Christophe Dejours, for example.
Response: Thank you very much for your valuable comment and for highlighting the connection between our findings and the concepts addressed in work psychology and psychodynamics. As you rightly mention, we acknowledge that work can simultaneously be a source of satisfaction and suffering due to the demands and challenges inherent in the role of school nurses across the various autonomous communities where they perform their duties. We will consider in future research introducing this point of view.
We especially appreciate your recommendation to review the works of Christophe Dejours, whose perspective could greatly enrich the interpretation of our results. We will review and incorporate these references into the next qualitative phase of our study, enabling us to delve deeper into the analysis of the impact of work demands on the satisfaction and professional quality of life of nurses.
Once again, thank you for your valuable insights.
- Still on page 12, the authors mentioned that "Nurses with 5 to 10 years of experience report lower professional quality of life, possibly due to unmet expectations and job realities that generate dissatisfaction. This finding is consistent with previous studies suggesting that less experienced nurses are generally more satisfied." Right, I do not dispute the validity of this statement, but what remains unsaid is that we also know that less experienced nurses have less room to express their dissatisfaction. In this regard, employment status is not irrelevant, of course, as we know that less experienced workers (often with more precarious contracts) tend to work longer hours and are frequently assigned the more difficult and/or more dangerous tasks. This reflection also serves to point out that this is a limitation of purely quantitative research, i.e., apparently, younger nurses have higher satisfaction levels. Fine, great, but do they have the "freedom" (the word may not be the best) to express their dissatisfaction? Perhaps qualitative research could help to understand this issue better. Once again, congratulations!
Response: We sincerely appreciate your thoughtful comment. You have highlighted an excellent point regarding the limitations of purely quantitative research in capturing the nuanced realities of less experienced nurses, including their ability (or lack thereof) to express dissatisfaction. We value your reflection on how employment status and precarious contracts can influence both their workload and their perceived levels of satisfaction.
In response to your comment, we have incorporated this reflection into the discussion section of the article (page 13), acknowledging it as an important consideration. Additionally, we will ensure that this perspective is further explored in the qualitative phase of our research, as it provides a valuable approach to better understanding the dynamics of professional quality of life among nurses.
Once again, we thank you for your thoughtful feedback and for pointing out an aspect that will undoubtedly enrich our future analysis.

Reviewer 2 Report
Comments and Suggestions for Authors
The paper reports a study evaluating the possible relationship between nurses' job satisfaction and the quality of professional life, presenting some results and conclusions that converge with other studies. The originality of the article lies in the fact that it situates the study in the context of school organizations in Spain, representing a contribution to the production of knowledge in this specific professional sector of nurses.
In general, the article complies with the standards, which is why it can be published (The paper is available in an online preprint).
Just a few small corrections to mention:
1) In the legend of Tables 1, 2, 3, 4, please identify the statistical method used to analyze the results.
2) In lines 213, 217, 219 and 416, insert a space between the period and the beginning letter of the following sentence. Please rectify;
3) In line 213, please, correct the plural of the acronym “NGO”. Acronyms do not have plurals.
4) In the final references...
4.1.) Please, remove capital letters from the names of papers, reserving the use of capital letters only for the first word.
4.2.) Please, write out the name of the journals mentioned in the references (for example, references 6, 8, 9, 10 and others).
4.3.) Please, remove the use of capital letters in the names of papers included in references 7 and 34.
4.4.) Please, if possible, insert the volume of the journal in reference 23.
Once these small corrections were made, the article could be published.
Author Response
Revisor 2
The paper reports a study evaluating the possible relationship between nurses' job satisfaction and the quality of professional life, presenting some results and conclusions that converge with other studies. The originality of the article lies in the fact that it situates the study in the context of school organizations in Spain, representing a contribution to the production of knowledge in this specific professional sector of nurses.
In general, the article complies with the standards, which is why it can be published (The paper is available in an online preprint).
Just a few small corrections to mention:
1) In the legend of Tables 1, 2, 3, 4, please identify the statistical method used to analyze the results.
2) In lines 213, 217, 219 and 416, insert a space between the period and the beginning letter of the following sentence. Please rectify;
3) In line 213, please, correct the plural of the acronym “NGO”. Acronyms do not have plurals.
4) In the final references...
4.1.) Please, remove capital letters from the names of papers, reserving the use of capital letters only for the first word.
4.2.) Please, write out the name of the journals mentioned in the references (for example, references 6, 8, 9, 10 and others).
4.3.) Please, remove the use of capital letters in the names of papers included in references 7 and 34.
4.4.) Please, if possible, insert the volume of the journal in reference 23.
Once these small corrections were made, the article could be published
Response:
Dear Reviewer,
We sincerely appreciate your valuable feedback and the thorough review of our manuscript. We have carefully considered your suggestions and made the following revisions to enhance the quality and clarity of our work.
In the legends of Tables 1, 2, 3, and 4, we have now identified and included the statistical methods used to analyze the results. Additionally, we have corrected the formatting by inserting spaces after periods in lines 213, 217, 219, and 416 to ensure proper sentence structure. In line 213 we have deleted the word NGO after reviewing the categories of contracting entities.
Regarding the final references, we have made several adjustments: capital letters have been removed from the titles of the papers, retaining capitalization only for the first word of each title. Furthermore, the full names of the journals have been written out in the references, including those in references 6, 8, 9, 10, and others, to provide complete citation information. In references 7 and 34, we have eliminated the use of capital letters in the article titles to maintain consistency with the citation style. Lastly, we have included the volume number of the journal in reference 23 to ensure the citation is complete.
We believe these revisions have significantly improved the clarity and quality of our manuscript. Thank you once again for your insightful comments and recommendations. Should you have any further suggestions or require additional information, please do not hesitate to contact us.
Sincerely

Reviewer 3 Report
Comments and Suggestions for Authors
1. Introduction section: clearly indicate the author's contribution to the research and define the scientific theory. Provide sources and links between the study and the scientific theory.
2. Discussion section: more closely compare the author's findings with those of other authors. Clearly indicate studies that confirm and do not confirm the phenomenon of job satisfaction and quality of professional life. It would be interesting to indicate differences in positions.
3. Complete the Discussion and References sections with the latest publications on the researched problem - 2022-2024.
Author Response
Response: Thank you very much for your valuable comments. Considering your feedback and that of the other reviewers, parts of the introduction have been modified, highlighting the theoretical model on which we have based our work and emphasizing the contributions this research can provide (page 2 lines 81-87).
In the discussion, the findings of the present study have been related to previous evidence on the topic, highlighting the relationship between professional quality of life and job satisfaction among school nurses (page 13: lines 448-475).
Finally, a new search was conducted on job satisfaction and professional quality of life among school nurses with the aim of finding out the latest publications on the researched problem. But no other references published between 2022 and 2024 were found. We would be most grateful if the reviewers knew any significant references relevant to our study, and could kindly inform us, as we will take them into consideration and include them in our manuscript.